# Paleoenvironmental Study of the Late Cretaceous–Eocene Tethyan Sea Associated with Phosphorite Deposits in Jordan

Mohammad Alqudah [1,*] , Nizar Abu-Jaber [2] , Abdulla Al-Rawabdeh [1,3] and Mahmoud Al-Tamimi [1]

1 Department of Earth and Environmental Sciences, Yarmouk University, Irbid 21163, Jordan
2 Department of Civil and Environmental Engineering, German Jordanian University, Amman 11180, Jordan
3 Laboratory of Applied Geoinformatics, Yarmouk University, Irbid 21163, Jordan
* Correspondence: mohammad.alqudah@yu.edu.jo

**Abstract:** Petrological, geochemical, and biostratigraphical investigations have been conducted on seventy-nine samples from four selected sections in Jordan to understand the factors that influenced the enrichment and deposition of massive phosphorite deposits. The calcareous nannofossil marker species *Broisonia parca constrica* and *Reticulofenestra bisecta*, from the assigned sections 1 and 2 (Hawar and Al Dhahikiyya), were indicative of the Campanian and Eocene periods, respectively. The enrichment of some ecological marker species such as *Kamptnerius magnificus* indicates that there were cold periods during phosphate precipitation. All thin sections of the phosphate samples are of grainstone and packstone textures and are composed of skeletal fragments and peloids. Skeletal fragments appeared to be the main component of Hawar phosphates with the existence of peloids, while peloids were the component in sections 2 and 3 (section 3: Al Hisa). At least three intervals of high phosphorous values appeared in the Hawar section, showing variations in the Ca and Nd isotopes and in the terrigenous inputs. Four periods of phosphate enrichment are observed in Sections 2 and 3. These are deep water circulation periods of the early and late Campanian period, interrupted by two periods of submarine and continental weathering. Deep water circulation was initiated during cooling in the Campanian period and indicated by high phosphorous and Ca isotope components and a decline in terrigenous indicators (Al, Si, Ti, and Fe). Submarine weathering during warmer deep-sea periods is indicated by a rising Nd isotope ratio when many of the igneous provinces were subjected to weathering. Continental weathering took place in the warmer periods, with the hydrologic cycle and enhancement of terrigenous indicators (Al, Si, Ti, and Fe) being observed. The effect of the hydrologic cycle was at its highest in the south during the Campanian period and in the Eocene, as both represented shallower settings.

**Keywords:** phosphates; depositional environment; Nd isotope; Ca isotope; petrography; biostratigraphy





## 1. Introduction

Massive phosphate deposits of economic interest occur in the southern margin of the Neo-Tethys. These deposits are one of the important natural resources in Jordan and the region [1]. Many prospective areas that contain considerable thicknesses of these deposits are distributed throughout southern Neo-Tethys. Still, despite these highly promising perspectives there are a range of unknowns and uncertainties, in particular with regard to the origin of these huge reserves.

Scientists have tried for a long time to understand the genesis of phosphates, but they struggled with the fact that phosphates can be found in divergent oceanic settings; some are found in the open marine system, e.g. [2], others are found in the infratidal to circatidal environments [3] and there is also phosphate deposition in estuaries and semi-restricted embayments [4,5]. These vast occurrences of phosphates make reaching a consensus on one model that explains the occurrences of phosphatic layers impossible.

Phosphates are extensive oceanic deposits that resulted from a series of interconnected tectonic, sedimentological, and oceanic conditions [6]. Each case can be studied separately based in differences in land–sea configurations which influence the oceanographic setting and the sedimentological processes. Adding to the previous point, the concentration of phosphorous in the water column, as well as climatic conditions and oxygen content, played major roles in the accumulation.

Phosphorite deposits in the ocean are associated primarily with the productivity of surface water that has been detected during the Late Cretaceous period in the southern Neo-Tethys [1,2,5,7]. Primary productivity is considered particularly in upwelling areas, where phosphate is released from the degradation of organic matter in the deep ocean. Upwelling current brought phosphorous and other nutrients to coastal areas and enhanced the primary productivity of these components in the water column [1,5].

Scientists have focused on the climate as an enrichment factor for phosphate deposits, as it seems to have been accelerated in warm humid climates as a result of a regional sea level rise that caused the establishment of mesotrophic conditions and the enhancement of continental weathering [2,8,9]. In such conditions, organic matter can be a source of authigenic phosphate through bio-chemical weathering and direct degradation in the ocean [3,6,8].

Phosphate formation requires suboxic and sometimes anoxic alkaline conditions. This can be in restricted basins or in the oxygen minimum zones, where organic matter degraded to produce phosphorus ions directly or from iron oxides [3–5,8,10,11].

The depositional environments of phosphates and oil shales have been investigated in Jordan, e.g., for phosphates: [1,12–14]. The close relation in the formation setting of both of them has been proposed by Ref. [1]. Oil shales have been investigated from 29 boreholes covering most of the oil shale deposits in Jordan [15–18]. The understanding of oil shale deposition went beyond the enrichment of nutrient and organic productivity during the upwelling; tectonisms and climate also played major roles in controlling the deposition [17]. Since phosphate is associated with the deposition of oil shales, the validity of an oil shale model in the deposition of phosphate was checked in this research work. Ref. [1] interrelated the phosphate deposits in Jordan with the tectonic obstruction, i.e., paleo-highs that prevented the upwelling of oceanic currents to pass through eastern Neo-Tethys which resulted in the deposition of phosphate deposits. The upwelling system, tectonisms, paleoclimates, terrigenous inputs, sedimentary environments, and the oceanographic factors must be hypothesized to come up with a comprehensive picture of the deposition of the phosphates.

This study aims to establish an age model in order to correlate phosphate deposits across vast areas. This model allows us to understand the paleo-oceanographic setting and climatic/tectonic conditions that interplay in the accumulations of phosphorite minerals. From one side, it includes the shallow and deep-water circulation of the Tethyan Sea, and from the other side, it demonstrates how the continent influences the enrichment of phosphates in the water column.

## 2. Geological Setting (from Late Cretaceous–Eocene)

The Cretaceous period comprised the warmest period in the Earth's history, and the Neo-Tethys of the late Cretaceous period flooded the inlands several times, resulting in widespread carbonate sedimentation [19]. The Campanian is considered to be the transition period between the harsh, warm mid-Cretaceous greenhouse and the relatively "cool" Maastrichtian and early Paleocene greenhouse [20–23].

Jordan was located on the southeastern part of the Neo-Tethys, which flooded the continental margins and widely opened eastwards [19,24,25]. In Jordan, chalk, chert, and phosphorites were formed in a pelagic or hemipelagic ramp environment [26]. Tectonically, the African–Arabian plate was in convergence with the Eurasian plate, resulting in an instability of the basin architectures. This caused diversity in the sedimentation pattern from shallow to deep deposits. Furthermore, the fluctuation in the eustatic sea level [27]

and the tectonic activities along the convergence boundary acted as control factors on the depositional patterns. Tectonic activities caused swells and basins [15,17,24,26,28]. The clockwise gyre circulation between Eurasia and Africa caused a westwards flow of deep, cold, nutrient-laden waters to the eastern African continental margin, which resulted in the enrichment of the water column by organic matter and phosphate [1,9,19,26,29,30].

Deep sedimentation continued into the Eocene period, during which the Arabian plate collided with the Eurasian plates [31]. This caused the termination of Neo-Tethyan flow during a period of major regression in the eastern margin of the African plate and a restriction of basins.

The two different staratigraphic units were investigated in this study: the Al Hisa Phosphorite Limestone (Campanian) and the Wadi Shallala Chalk (Eocene) formations. sections 1, 3, and 4 (Hawar, Al Hisa, and Al-Shydiyya) belong to the Al Hisa Phosphorite Limestone Formation. This formation is composed mainly of phosphate layers and is distributed in north (Hawar), central (Al Hisa), and south (Al-Shydiyya) Jordan. The overall trend is thinning to the north and south from central Jordan [24]. It is well known that this formation was deposited on the inner part of the broad shelf [26].

Section 2 (Al Dhahikiyya) is an Eocene succession located in the east of Jordan which stratigraphically belongs to the Wadi Shallala Chalk Formation. The Wadi Shallala Formation is deposited in a deep setting in the north of Jordan and in a shallow inner part of a narrow shelf in the eastern part of Jordan, and is composed of chalk and chalky limestone [32].

## 3. Material and Methods

### 3.1. Studied Sections

A total of 79 samples were collected from 4 sites: section 1 (Hawar), section 2 (Al Dhahikiyya), section 3 (Al Hisa), and section 4 (Al-Shydiyya). The following are the detailed sampling and lithological descriptions of the different sections. All samples and their generic descriptions are summarized in Table 1.

Section 1: A total of 29 samples were taken from the Amman Silicified Limestone in the Hawar Area (Figures 1 and 2). The samples represented the chalk, chert, and phosphate intervals. Three units are recognized in the section. At the bottom, chalk appears and is imbedded with thin gray chert bands. The thickness of the cherts increases upward (Figure 3). The cherts become dominated in the second unit. Silicified chalk and chalky limestone are present between the chert layers. The chert is replaced by phosphate and phosphatic limestone in the third unit. The phosphate becomes dominant with thicknesses reaching three meters. Phosphate layers are composed of hard and massive granular phosphatic limestone. One and half meters of massive white chalk, occasionally laminated, interbedded with dark fossiliferous limestones, characterizes the bottom of the section. Well-preserved macrofossils such as gastropods and bivalves ranging between 2 and 3 cm appear in the fossiliferous limestone layer. Chalk grades upwards to gray and contains well-preserved fossils. Various colors of chert bands lie at the top of the massive chalk layer. Occasionally, light- and dark-gray thick chert beds with imprecations of silicified macrofossils are observed at 30 cm and some are half a meter thick per layer. Chalk is interbedded between the chert layers and is characterized by its hardness due to the silicification. Chalk and chalky limestones become silicified and contain well-preserved shark teeth, macrofossils, and parts of marine animal skeletons. About two meters of limestone bed, including limestone nodules, overlies the chert beds. Fossils are less abundant, and the number of chert bands decreases as we move upwards. This layer is well characterized by intensive silicification, which causes the high hardness of these strata. Limestone grades into chalk as we move upward. The base of the chalk layer is of a white color, and it gradually changes into pale yellow due to existence of a granular texture. Hard white phosphatic limestone consists of white granules on a grey background and lies at the top of the chalk layer. A few chert bands appear between the phosphatic limestone layers. A round 3 m of phosphates rest at the phosphatic limestone layer, it is very hard and is

composed of white to yellow granules set within a very grey hard matrix. Bone fragments and shark teeth appear in this layer. Phosphate grades upwards to chalky phosphate and becomes softer. At the top, yellow marly chalk appears, containing clay minerals and an absence of macrofossils.

**Table 1.** Samples and the related analysis in this study.

| Sample | Description | Age Determination | Petrography | Isotope | Elements |
|---|---|---|---|---|---|
| H1-4 | Chalky limestone | X | | | |
| H5-10 | Silicified limestone | X | | | |
| H11 | Silicified limestone | X | X | | |
| H12-15 | Hard silicified phosphate | X | | X | X |
| H16 | Hard silicified phosphate | X | X | X | X |
| H17 | Hard silicified phosphate | X | X | | |
| H18-21 | Phosphate | X | | X | X |
| H22 | Phosphate | X | X | X | X |
| H23-27 | Phosphate | X | | X | X |
| H28 | Marly chalk | X | X | | |
| H29 | Marly chalk | X | | | |
| D1-3 | Bituminous chalk | X | | | |
| D4-6 | Siliceous chalk | X | | | |
| D7 | Siliceous chalk | X | | X | X |
| D8 | Siliceous chalk | X | X | X | X |
| D9 | Glauconite-bearing chalk | X | | X | X |
| D10 | Glauconite-bearing chalk | X | | X | X |
| D11 | Phosphatic-glauconitic arenite | X | | | |
| D12 | Phosphatic-glauconitic arenite | X | X | X | X |
| D13 | Phosphatic-glauconitic arenite | X | | X | X |
| D14 | Phosphatic-glauconitic arenite | X | X | X | X |
| D15 | Phosphatic-glauconitic arenite | X | | | |
| D16 | Phosphatic-glauconitic arenite | X | X | X | X |
| D17 | Chalk | X | | | |
| D18-19 | Chalk | X | | X | X |
| D20 | Chalk | X | X | X | X |
| D21 | Chalk | X | | | |
| HS1-2 | Coquina layer | | | | |
| HS3 | Phosphate | | | X | X |
| HS4 | Phosphate | | X | X | X |
| HS5-6 | Phosphate | | | X | X |
| HS7 | Phosphatic marly limestone | | | X | X |
| HS8 | Phosphate | | X | X | X |
| HS9-12 | Phosphatic marly limestone | | | X | X |
| HS13 | Phosphate | | | X | X |
| HS14 | Phosphatic marly limestone | | X | X | X |
| HS15-16 | Phosphatic marly limestone | | | X | X |
| HS17 | Phosphatic marly limestone | | | | |
| HS18 | Phosphate | | | | |
| HS19 | Phosphatic marly limestone | | | | |
| HS20 | Phosphate | | X | | |
| HS21 | Phosphatic marly limestone | | | | |
| SH1 | Sand | | | | |
| SH2 | Calcareous phosphate | | | | |
| SH3-4 | Sandy limestone | | | | |
| SH5 | Calcareous phosphate | | | | |
| SH6-7 | Sandy limestone | | | | |
| SH8 | Calcareous phosphate | | | | |

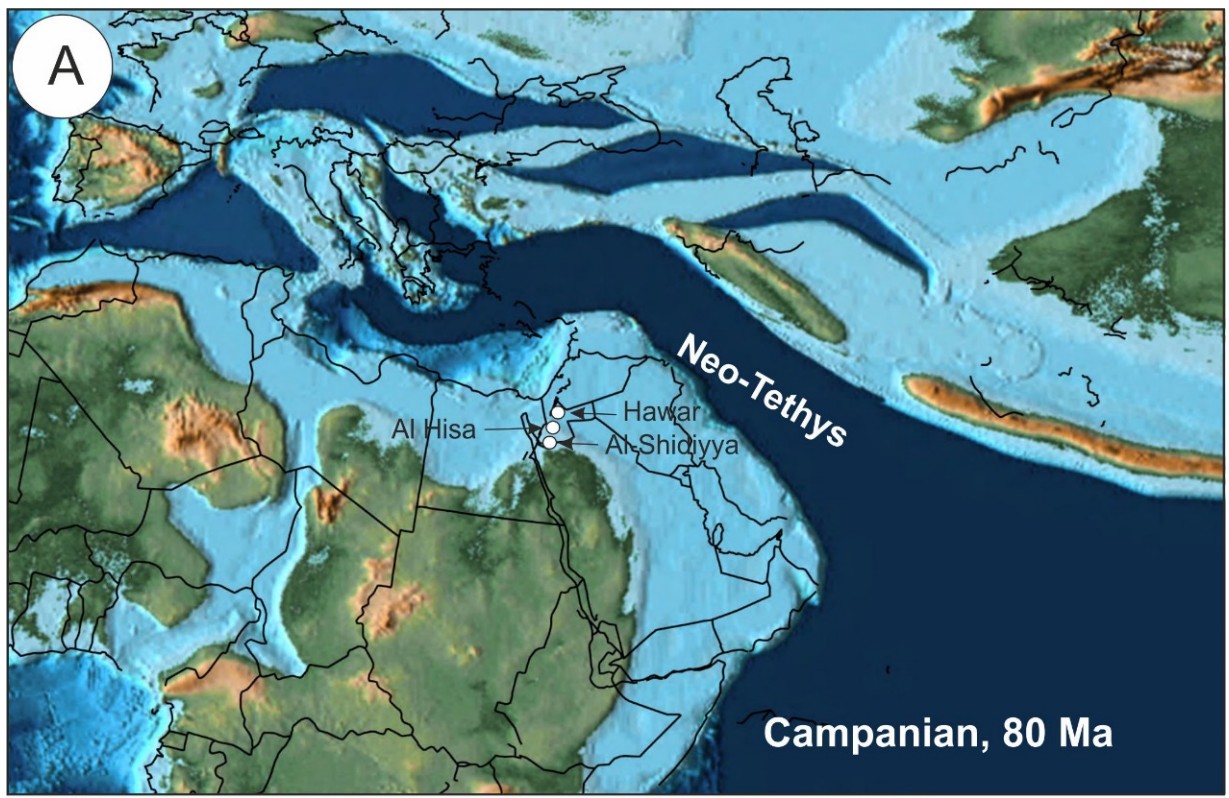

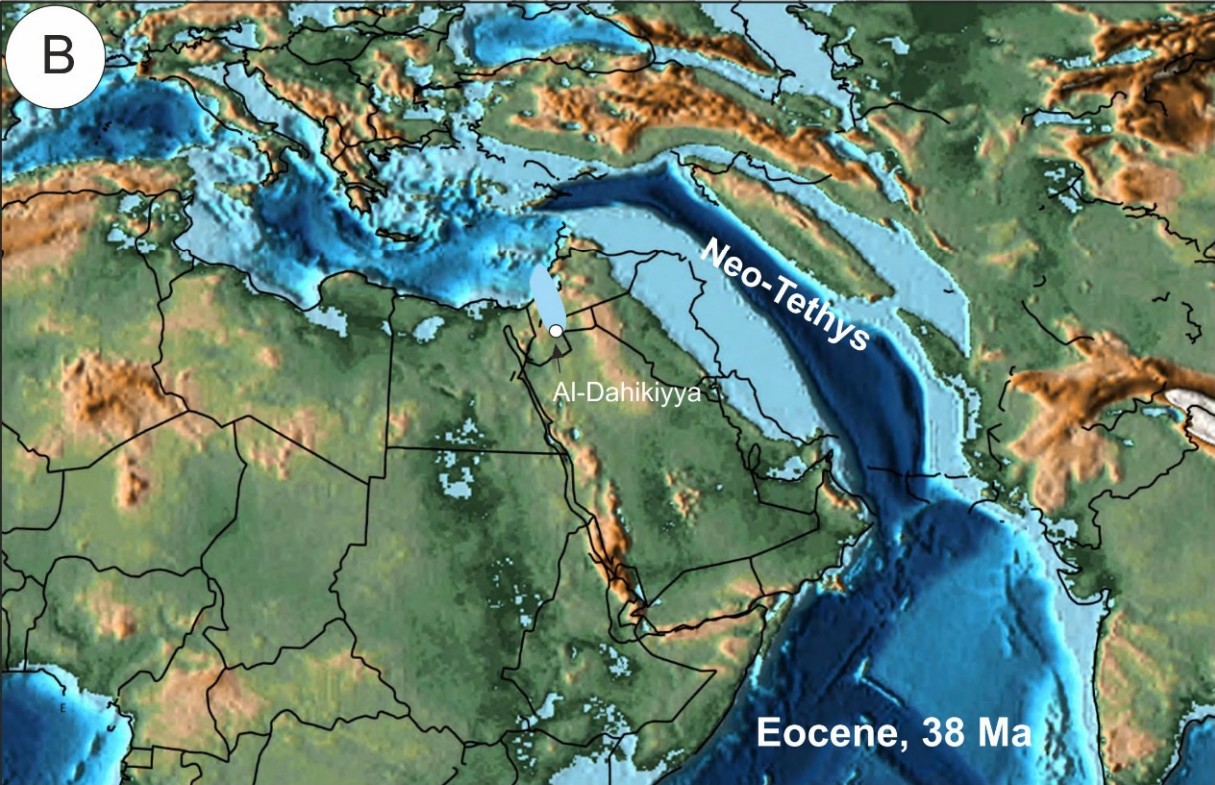

**Figure 1.** Paleogeographic maps of Jordan and Neo-Tethys during the Campanian and Eocene periods: (**A**) location position of HawarAl-Hisa and Al-Shidiyya sections in the southern Neo-Tethys showed that deposition of phosphate occurred in the epicontinental sea during Campanian period; and (**B**) location position of Al-Dhahikiyya during Eocene period represents the gateway from Al-Dhahikiyya into Neo-Tethys (modified after Refs. [33,34]).

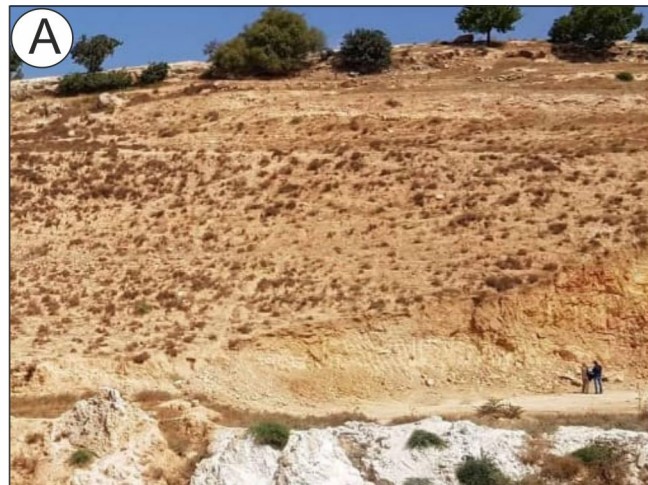

Hawar Section

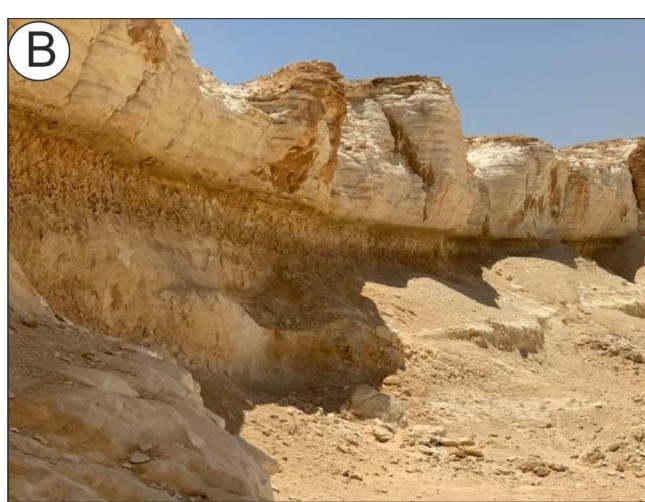

Al-Dhahikiyya Section

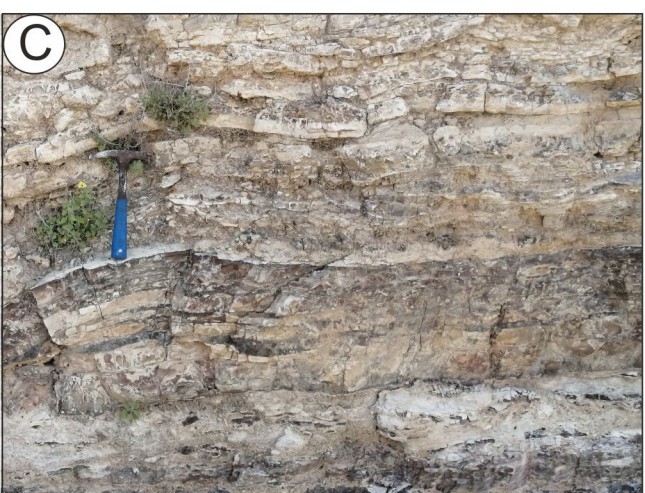

Al-Hisa Section

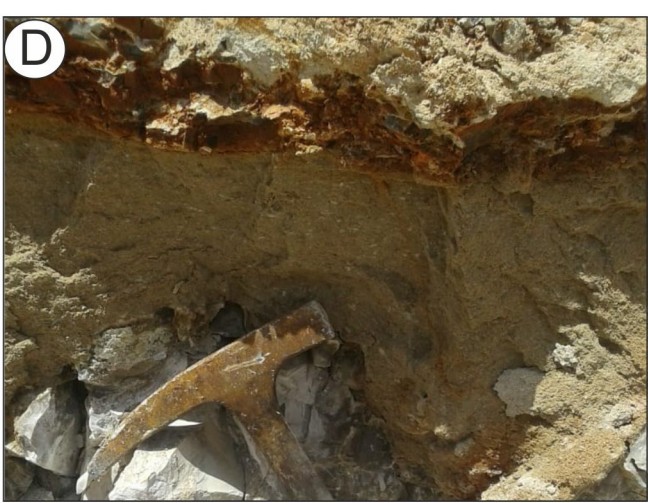

Al-Shidiyya Section

**Figure 2.** A field photos of (**A**) the lower part of section 1, (**B**) section 2, (**C**) the main bed in section 3, (**D**) The A3 horizon in section 4.

Section 2: A total of 21 samples were collected from the bituminous chalk layer, veined chalk, phosphatic arenite, and the overlying chalk layers at Al-Dhahikiya (Figure 3). Three meters of dark bituminous chalk were found at the bottom. Siliceous chalk including nodules of barite "desert roses" overlies the bituminous chalk. Glauconite-bearing chalk appears with the increasing glauconite as we move upward and a layer of phosphatic-glauconitic arenite overlies it. Chalk appears again at the top of the section (Figure 3). This section is of the Eocene age, therefore it cannot be correlated to the other sections.

Section 3: A total of 21 samples were taken from the marly limestones and phosphate layers at the Al-Hisa area. Five phosphate layers were recognized in the Al-Hisa mine (Figure 2C) and are bottomed with a coquina layer. The coquina layer mainly includes an accumulation of well-preserved *Lopha villi*. The coquina layer was not found in section 1. However, a round three-meter phosphate layer overlies the coquina. The five phosphate layers appear in various thicknesses and are intercalated with marly limestones, limestones, siliceous limestones, and cherts. It appears that number of phosphate layers in this section are higher than in section 1.

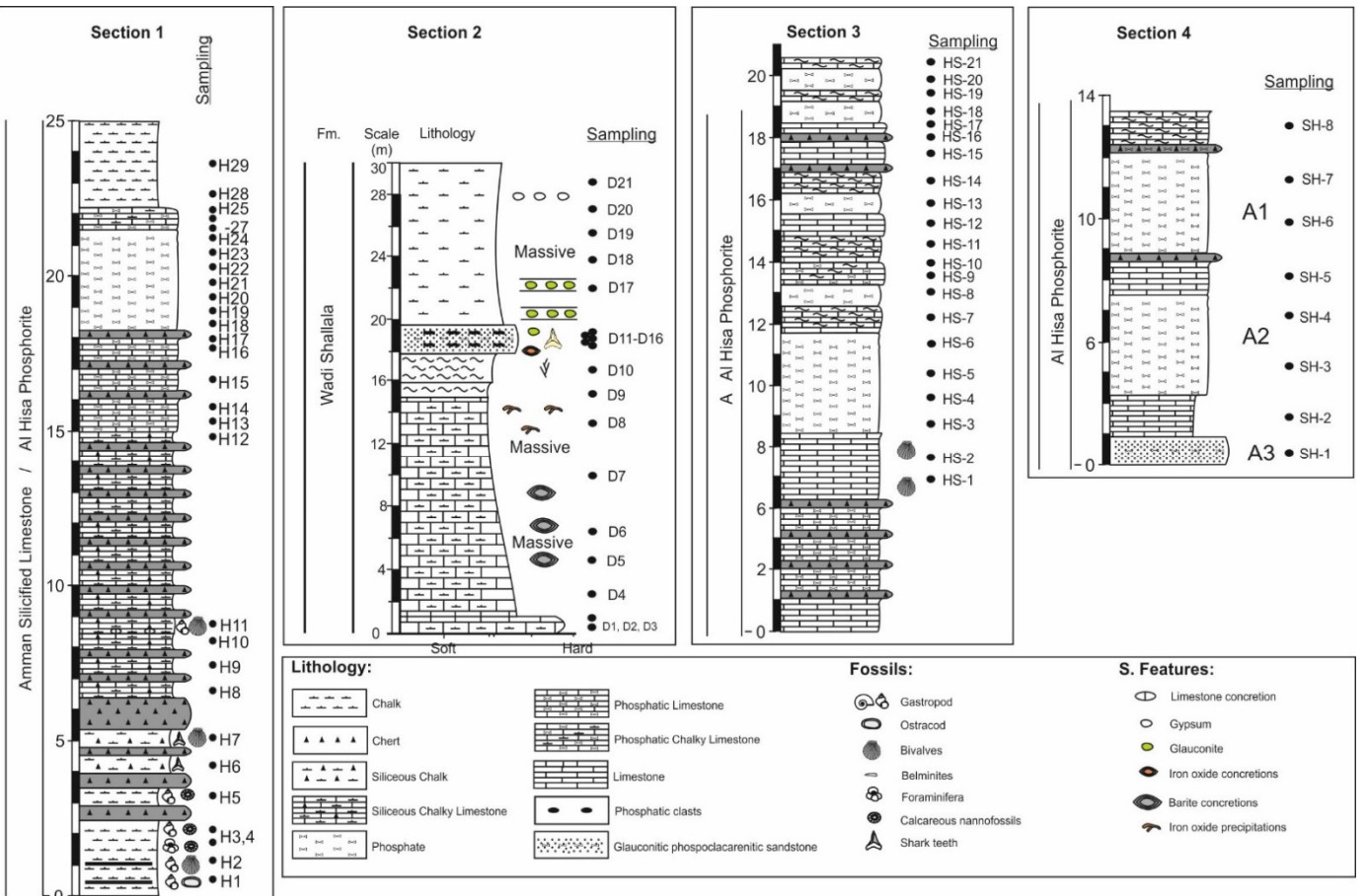

**Figure 3.** Columnar sections show the main lithological and fossil contents in sections 1–4.

Section 4: Eight samples were taken from the three phosphate horizons A1-3 and from the layers between WA-1 to 3 in Al-Shydiyya. Three distinctive horizons of sandy phosphates appear, well known in the literature as A1, A2, and A3, and separated by three silicified phosphatic limestone horizons that are given the abbreviations of WA-1 to WA-3.

### 3.2. Micropaleontological Analysis

Micropaleontological, petrological, geochemical, and isotope data were obtained in this study. Calcareous nannofossil assemblages were investigated for a better understanding of the time framework of these depositions. Twenty-nine smear slides were prepared from section 1 and eighteen smear slides from section 2. Calcareous nannofossil assemblages were examined under a polarized microscope of 1500× magnification located in the Department of Earth and Environmental Sciences, Yarmouk University. Ref. [35] biozonation was used for the Campanian assignment and Ref. [36] was used for the Eocene biostratigraphy.

### 3.3. Petrographical Analysis

The petrographic study was performed by using a polarizing optical microscope with 25,100× magnifications. Thin and polished sections were prepared for detailed petrographic study in Yarmouk University's Geology Workshop. Phosphate components were investigated and recorded and then interpreted in terms of their depositional environment.

### 3.4. Geochemistry and Isotopic Analysis

Representative samples of the phosphate rock were chemically analyzed to identify their major and trace element composition. MC-ICP-MS (two multi-collector ICP-MS

instrument) located at ALS Scandinavia AB Lab was used for high precision isotope ratio measurement to characterize the collected samples. Ratios are reported with absolute and relative standard deviations down to 0.001%. Prior to measuring the isotopic composition, samples underwent a digestion process following Ref. [37]. $Fe_2O_3$, $CaO$, $P_2O_5$, $SiO_2$, $Al_2O_3$, and $TiO_2$ were plotted to understand the different parameters affecting the precipitation of phosphates. Ratios of $^{44}Ca/^{42}Ca$ and $^{143}Nd/^{144}Nd$ in carbonate fluorapatite can be an indication of the intensity of phosphogenesis and describe the long-term changes in ocean circulation and continental weathering [9]. The Nd isotopic compositions ($^{143}Nd/^{144}Nd$) may help to decipher rates of phosphate deposition [9]. The $^{143}Nd/^{144}Nd$ ratio in rocks depends upon their initial $Sm/Nd$ ratio and their age, given that the radiogenic isotope $^{143}Nd$ is produced by the decay of $^{147}Sm$. Rare earth elements (REE) are preferentially incorporated into the matrices of apatite crystals, which explains the depositional conditions and post depositional alterations of phosphatic sediments [38].

## 4. Results

### 4.1. Biostratigraphy

Calcareous nannofossil assemblages are rich in samples H1-4 at the bottom of the section and in samples H28-29 at the top. The appearance of the calcareous nannofossil marker species *L. cayeuxii* and the disappearance of *L. septenarius* suggest that the bottom of section 1 should be assigned to the biozone UC-12 of the Santonian Age (Figure 4). Meanwhile, the top of section 1 is marked with the appearance of *Broisonia parca constrica*, which assigns the top of the section to the biozone UC-14/early Campanian (Figure 4). Biostratigraphy of section 2 is performed using only one marker species. *Reticulofenestra* dominates in the studied samples of section 2 (Figure 4). Appearances of the marker species *Reticulofenestra bisecta*, particularly in samples D-1 to D-18, suggest the biozone NP-17/late Eocene for section 2. Additionally, the enrichment of section 1 samples with some ecological species such as *Kamptnerius magnificus* associates the samples of H30-33 with cold periods. This means that cold periods characterized the end of the phosphate precipitations.

### 4.2. Petrography

Section 1 is characterized by three main facies: packstone, grainstone, and wackstone facies as shown in Figure 5A–E. The bottom of section 1 is represented by samples H1-11, which show the packstone facies with countable amount of bioclasts. Bioclasts are bone fragments, ostracodes, and planktonic and benthonic foraminifera. Many non-bioclast grains are found in the thin section of H11. Textural features in the phosphate samples (samples H12-22) represented mainly the grainstone facies (Figure 5B–D). All samples in this facies showed a high allochem percentage in comparison to the matrix. Many allochem types were found in the thin sections, such as peloids, intraclasts, and bioallochems, which included skeletons and shells. Peloids were oval-shaped, uncoated, and poorly sorted as there are three or more sizes of these peloids. Intraclasts had many irregular, poorly sorted, and angular shapes which indicated the in situ origin of these grains. Bone fragments and skeletons are well preserved and relatively large in size. Many of these skeletons contain pores within their structure which could indicate that these skeletons were deposited under calm conditions. The matrix in the middle zone is less abundant than it is in the bottom zone as grains dominate in this facies. Samples H23-32 obtained the wackstone facies as the calcitic matrix is dominant. Bone fragments (Figure 5E) are the major bioallochem component in sample H28. These components are well preserved and contain voids within their structure.

Section 2 samples are characterized by three main facies: wack-mudstone, and two different sub-types of grainstone facies, these facies are shown in Figure 5F–J. The bottom and top of these sections are characterized by planktonic foraminifera supported with a calcitic matrix (Figure 5F,G,J). This facies is wack-mudstone, which inferred that the deposition was pronounced in a deep setting. In between this facies, two different facies appeared in Al-Dhahikiyya, both are grainstones with grain support, but the difference

comes from the type of grain in the texture. The first one comprises pellets and intraclasts; peloids have an oval shape and are uncoated by external materials. Intraclasts have many irregular, poorly sorted, and angular shapes which indicate the in situ origin of these grains (Figure 3H). The second grainstone facies has less pelletal fractions and its intraclasts are coated with phosphatic and calcitic material (Figure 3H). These intraclasts are sub-rounded to rounded and sub-angular grains. Both grainstone facies are lacking the skeletal bioallochem component.

A variety of facies types were observed in section 3, ranging from grainstone to wackestone facies (Figure 5K–P). All section 3 samples showed the grainstone facies, except for samples HS 14-17 which represented the wackestone facies (Figure 5M). The base is characterized by fossiliferous limestone where *Lopha* spp dominated. Above the base, grainstone composed mainly of pelletal grains and the skeletal components as a minor constitute (Figure 5K) was observed. Skeletal grains were not found in any of the above sections. Coated and sometimes uncoated peloids supported with calcite cements were common in samples HS 4-13 (Figure 5L). Sorting appeared to be good to moderate in these samples. At the top, another phoshatic horizon represents well-sorted, uncoated peloids supported with calcitic cement (Figure 5K). Cement was not found in the top samples as the phosphate became friable (Figure 5L).

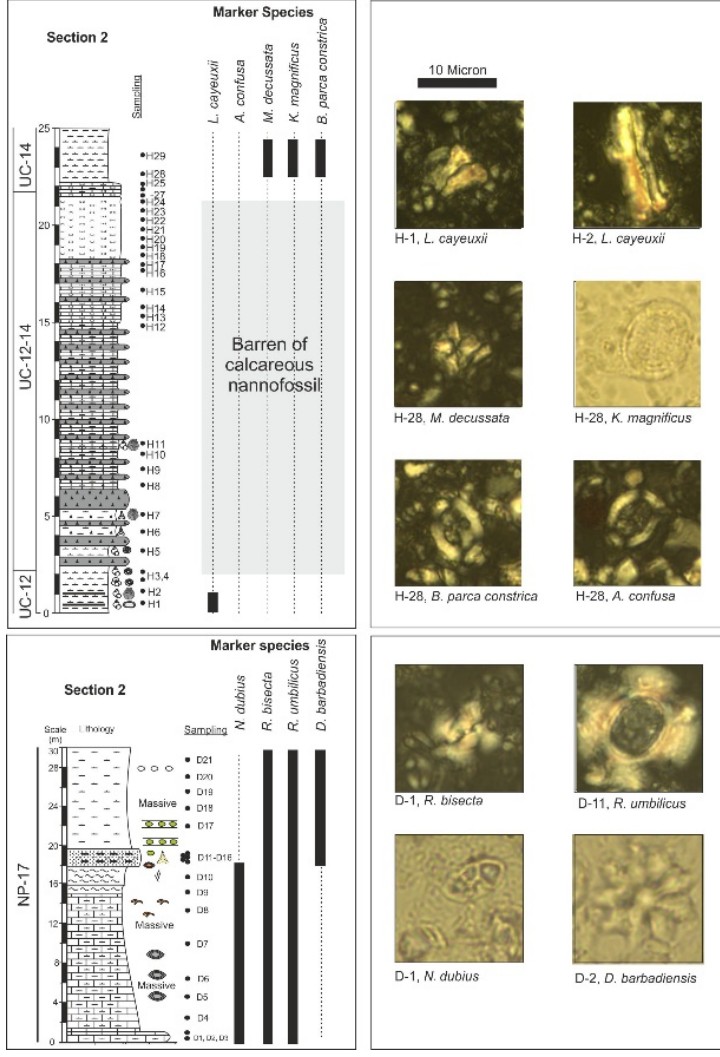

**Figure 4.** Distribution charts of marker species of calcareous nannofossils in sections 1 and 2. *L. Cayeuxii*, the marker species of biozone UC-12, *B. parca constrica*, the marker species for UC 14, and *R. bisecta*, the marker species for NP17 are illustrated in the plates.

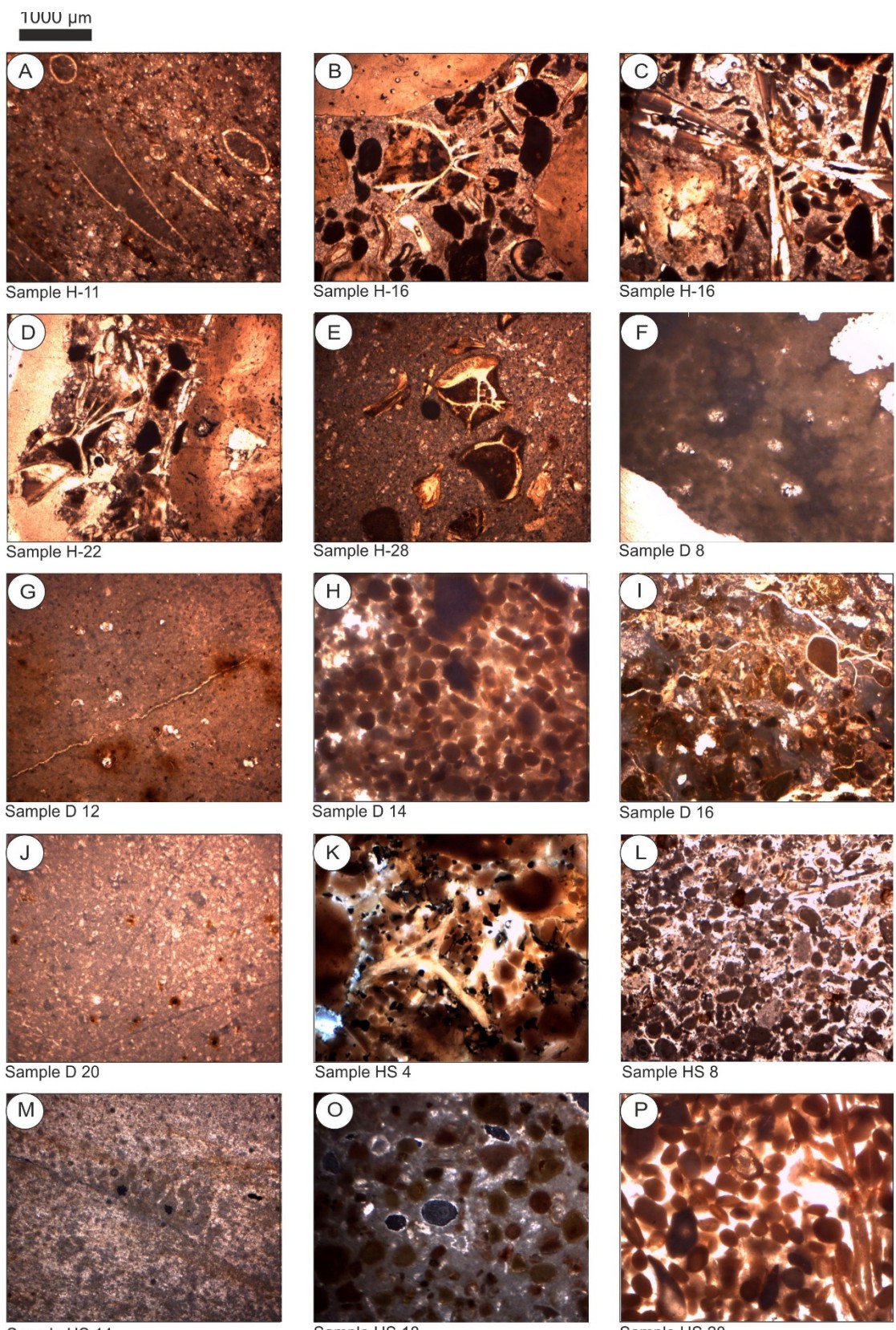

**Figure 5.** Microphotographs represent the textural contents of sections 1, 2, and 3. Microphotographs (**A,F,G,J,M**) show the wackstone facies. (**B–E,K**) represent the skeletal frames within the grainstone, while (**H,I,L,O,P**) show the pelletal components within the grainstone facies.

### 4.3. Geochemical Data

Three intervals of high phosphorous values appeared in section 1, the lower one (sample H1) was mixed with chalks and was indicated by high Ca values. In this horizon, there are observable high Fe, Al, Ti, and Al values associated with the high phosphorous values (Figure 6). The isotopic ratio εNd showed high values, and reversely in the δ43/42 Ca ratio. The second interval that represents high phosphorous in the Hawar section showed quite low values of Fe, Al, Si, and Ti (samples H4, 5, and 6) where the εNd ratio was and δ43/42 Ca was high. The top interval (samples H8, 9, 10, 11, and 12) represented high phosphorous values which were associated with a low εNd ratio and a high δ43/42 Ca ratio. Geochemically, high Al, Si, Fe, Si, and Ti values only showed anomalies at sample H11, and these elements do not show any value in samples H3, 8, 9, 10, and 12.

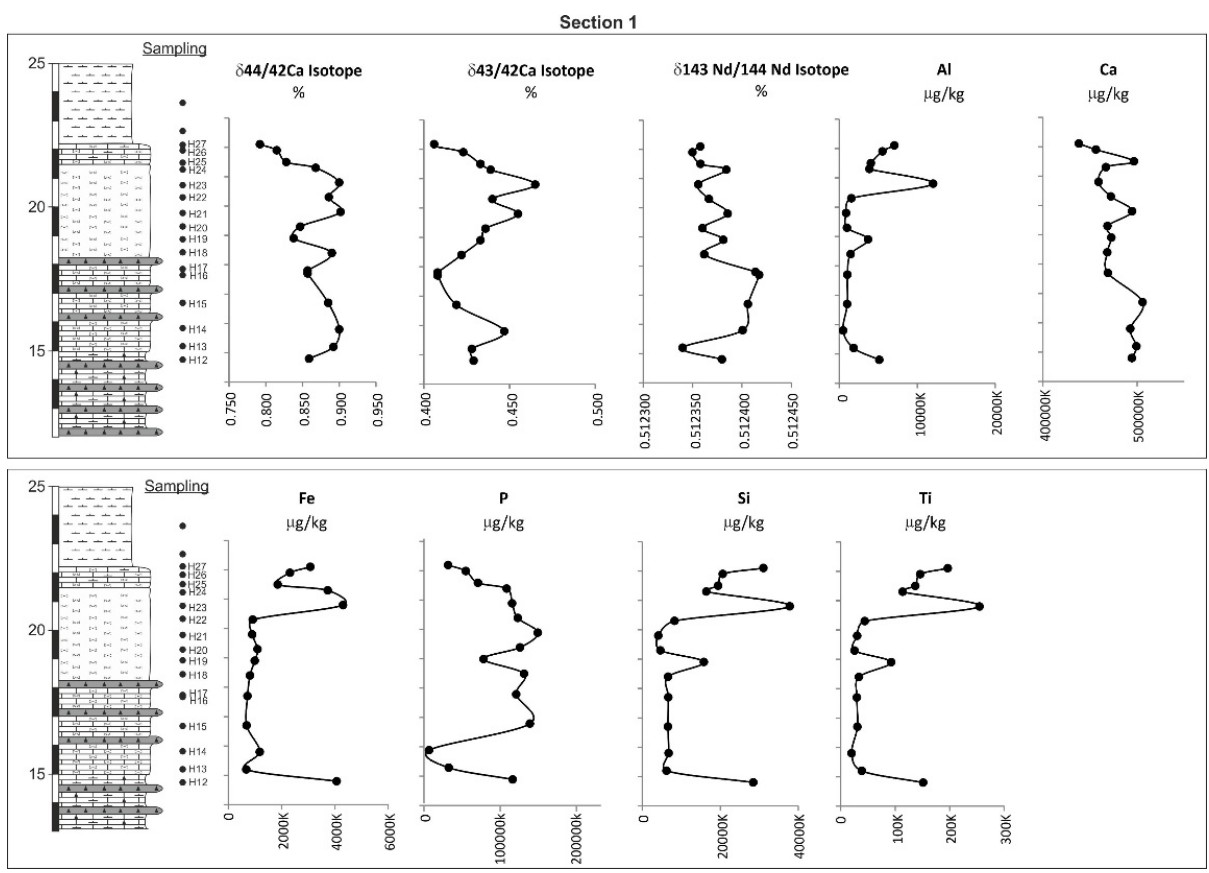

**Figure 6.** Geochemical data plot against its stratigraphic positions of samples 1.

Oil shale was recognized at the base of section 2 (sample D7 and 8). It was characterized by low Al, Si, Fe, Ti, and P. The isotopic ratio δ143/144 Nd was high, and the δ43/42 Ca ratio was high too. The first phosphate horizon appeared in samples D8 and 9 in combination with a carbonate background; in this horizon, there was significant increase in Al, Fe, Si, and Ti (Figure 7). The δ143/144 Nd showed a slight increase and a decline in the δ43/42 Ca isotopic signature. During the main regression stage of the sea, which is represented in samples D11-16, phosphorous showed high values and was accompanied with high Al, Si, Fe, and Ti. The δ143/144 Nd and the δ43/42 Ca isotope are in their minimum values. After a major transgression, the retrieved carbonate system indicated high Ca values in samples D18-21. In this interval, the third enrichment of phosphorous was detected with low values of Al, Si, Fe, and Ti and a slight increase in the δ43/42 Ca isotope.

**Figure 7.** Geochemical data plot against its stratigraphic positions of section 2 samples.

A phosphatic carbonate succession appeared at the base of section 3 (samples HS 3, 4, 5, and 6) represented by high Ca and P and a decline in Al, Fe, Si, and Ti values (Figure 8). Those values were correlated to the petrographic observation that bone fragments were enriched in this horizon, indicating that skeletal phosphate was present in the early stage of phosphate precipitation. Samples HS 7, 8, 9, 10, 11, and 12 showed sequences of two events: First, there is a relative drop in phosphorous associated with high Al, Fe, Si and Ti values, which could indicate that the terrigenous input acted as a dilution factor for the concentration of phosphorous in the water column. Second, high phosphorous values were combined with low Al, Fe, Si, and Ti values and relatively higher Ca values. In both events, the $\delta$143/144 Nd ratio was relatively high, while the $\delta$43/42 Ca isotope was low. The top phosphate horizon, similar to the bottom, was characterized by high Ca values, low Al, Fe, Si, and Ti values, and a relatively higher $\delta$43/42 Ca ratio. Above, the influx of terrigenous input, indicated with high Al, Fe, Si, and Ti values, characterized the top of the section where phosphorous disappeared in the water column.

The REE patterns, normalized to post-Archean Australian Shale (PAAS; Ref. [39]) are shown in Figure 9.

The REE patterns show consistent negative Ce anomalies in the analyzed samples, reflecting similar results from previous studies [40,41].

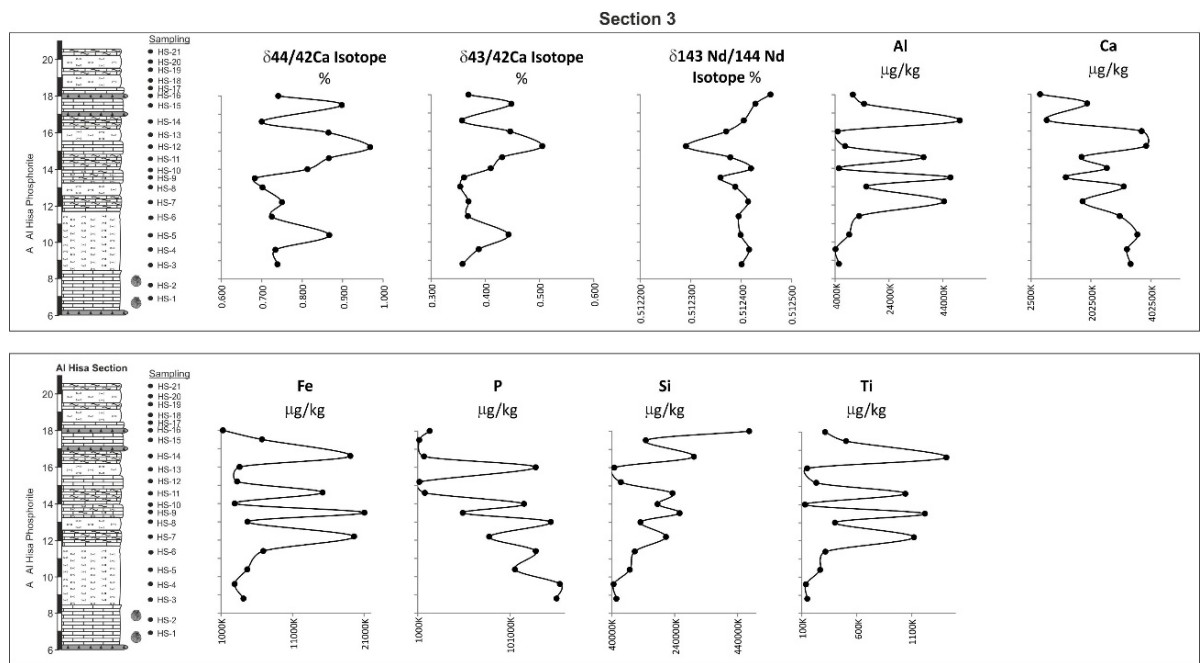

**Figure 8.** Geochemical data plot against its stratigraphic positions of section 3 samples.

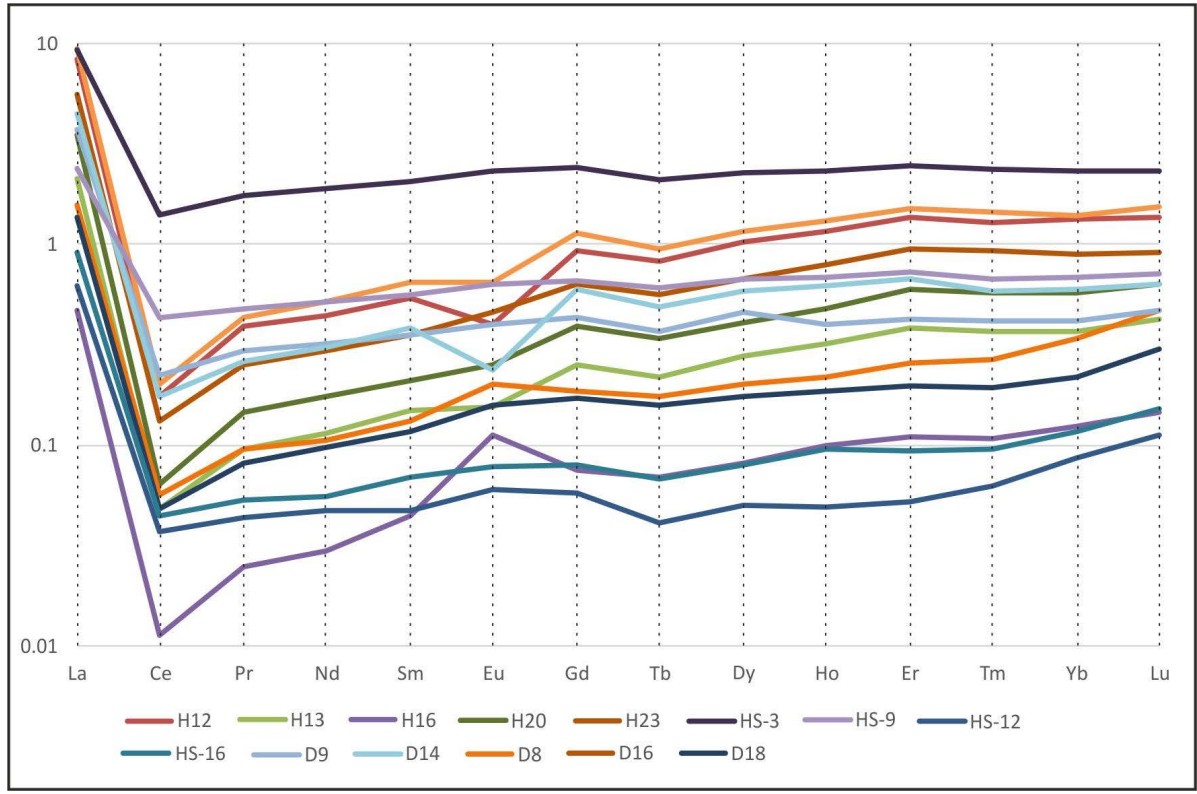

**Figure 9.** Rare earth element patterns normalized to post-Archaen Australian Shale (PAAS). Plotted samples were chosen to show the general pattern obtained from plotting the rare earth elements of the samples.

## 5. Discussion

### 5.1. The Distribution and Correlation of Phosphates in Jordan

Huge phosphate deposits in the eastern Neo-Tethys witnessed new oceanic and climatic conditions, facilitating an accumulation of a commercial quantity of phosphates

distributed all over Jordan. Long periods of production in Campanian phosphates have led to huge numbers of investigations focused on chemical and textural compositions, e.g., Ref. [30] for central Jordan; and Ref. [13] for north of Jordan. As complementary to those works, this study established a clear correlation between the phosphate-bearing strata between north, central, and south Jordan.

The phosphate in the north, represented by section 1, is different from central and southern Jordan (sections 3 and 4) in a number of respects. The onset of phosphate deposition was characterized by coquina layers in south and central Jordan [24,28], while they were absent in section 1 in the north. The proximity of south and central Jordan to the shoreline may have been an important factor in the forming of bioherms.

Textural differences were detected in this study. Grainstones of the studied phosphates can be divided into four categories; pelbiomicrite, the main facies in section 1; pelmicrite which exists in section 2; pelsparite in section 3; and loose peloidal grainstones. This suggests that bio-components were thriving in the distal parts of the basins.

*5.2. Sedimentary Environments of the Phosphates in Jordan*

Phosphates were deposited during the Campanian period in the middle to inner parts of the broad shelf in Jordan [24,26]. From a paleoenvironmental point of view, grainstone facies, which exist at the middle of section 1, were laid between the wackstone facies at the bottom and top. This has been confirmed by number of researchers, such as Ref. [13]. The same observation can be concluded in section 2 as described as well by Ref. [28]. Both cases indicated that the restriction and shallowing of the basin took place during phosphate precipitation. However, the types of grains brought more information on water depth in sections 1 and 2. The appearance of well-preserved skeletal grains, with voids in the samples and the absence of coatings on the peloids in section 1, indicated a calm, deep basin during the precipitation of phosphate for the period of the Campanian times. Meanwhile, in sections 2 and 3, the absence of skeletal grains and the existence of coated peloids support the idea that phosphate precipitation occurs in very shallow and high-energic environments which, in turn, supports the idea proposed by Ref. [24], that phosphates deposited in the inner parts of the shelf extend throughout Jordan. However, each phosphate deposition was affected significantly by its position in the shelf.

*5.3. The Controlling Factors for the Formation of Phosphates*

The productivity of phosphates in water columns and the deposition of phosphorite all over the world have been described in detail, e.g., [1]. Ref. [1] postulated that phosphates have been enriched in the Middle East through pulses of upwelling that have been obstructed by swells created by the tectonic activity of the late Cretaceous period. In this study, the factors that influenced the deposition of phosphate, including deep-ocean circulation and terrigenous influxes, have been discussed in detail.

5.3.1. Early Campanian Neo-Tethys Ocean

The oceanic setting of Neo-Tethys was significantly influenced by the instability of the plate tectonic during the Campanian period [19,42,43]. The continent land configuration which has resulted from the convergence of the African and Eurasian plates affected the oceanographic setting by closure westward and the opening of Neo-Tethys eastward [19]. It has been postulated that serpentinization, which typically is associated with continental conversion, releases significant volumes of phosphate [44]. Three pulses of cold-water bodies coming from the eastern Neo-Tethys was proposed by Ref. [45]. This is based on the dominance of cold-water species, such as *Arkhangelskiella confusus*, *Eiffellithus turriseiffelii*, and *Zeugrhabdotus erectus*, in the three intervals of the Campanian chalk in Jordan. These are associated with high nutrient species, such as *Arkhangelskiella confusus* and *Zeugrhabdotus erectus,* which indicated the preference of these species to a low paleotemperature and a high nutrient regime (Figure 9). This event was correlated to the Global Curriculum Current which has been reported previously by Refs. [1,5,29] and others. In this particular

period, surface and deep-water paleotemperature progressively declined [22]. Ref. [22] recorded the warming in the earliest Campanian period, but surface water experienced cooling afterwards which could have come from the deep oceanic fluxes (Figure 10).

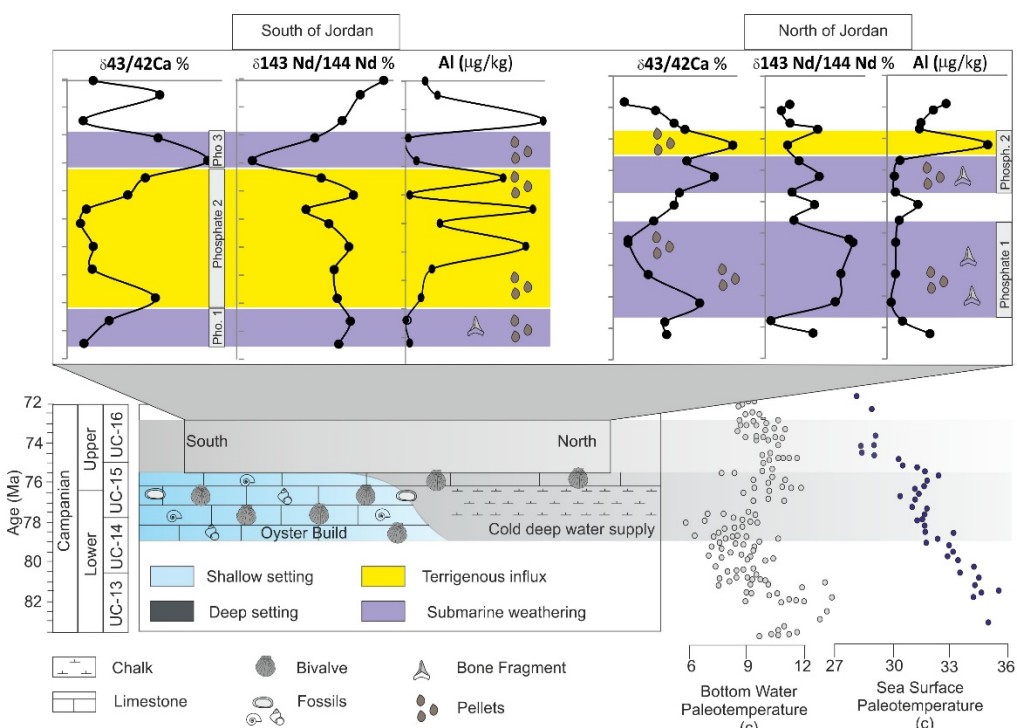

**Figure 10.** Integration of the geochemical, biostratigraphical, and petrographical interpretations shows the variation in oceanic circulation and continental weathering. Bottom and surface paleotemperature are taken from Linnert et al., 2014.

According to Ref. [22], sea surface temperature and deep-sea temperature at the end of this period showed a dramatic rise. In this circumstance, terrigenous influxes were recognized. High Fe, Al, Ti, and Al contents were recorded with the high phosphorous values indicating that nutrients were not only provided by deep oceanic currents, but also by influxes from hinterlands in a warmer period due to the enhancement of the hydrologic cycle which influenced the enrichment of phosphorous during the Campanian period.

Stable calcium isotopes show a light excursion in the south and a positive one in the north during this period of terrigenous input. Modern biomineralization tends to result in negative calcium isotope signatures [46]. This suggests that the calcium fixed into the apatite structure was incorporated inorganically.

Thus, the petrographic and geochemical data indicate that there was a clear difference in the depositional environment between central and north Jordan. An accumulation of the bivalve bioherms was clearly recognized at section 3, while chalk was dominant at section 1, even though terrigenous input seems to be at the base of section 1.

### 5.3.2. Submarine Weathering of Campanian

The oldest phosphate layer of both sections 1 and 3 showed similar petrographic and geochemical observations. Both have bone fragments and pellets in their apatite fraction. Geochemically, the terrigenous indicators were minimal and the Nd isotope was relatively high (Figure 10). This event was correlated to the high temperature in the deep-sea records of Ref. [22], indicating that the warming of the deep sea and submarine weathering took place during this warming period (Figure 11). Submarine pluming has been evidenced by a positive shift in the Nd isotope, a negative shift in the Ca isotope, and an absence of terrigenous input.

Stage 1: Submarine Weathering

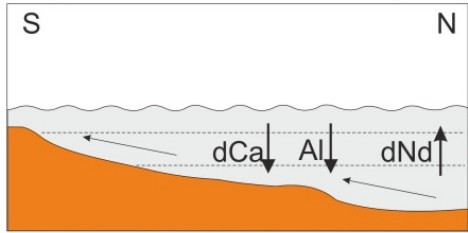

Stage 2: Continental Weathering

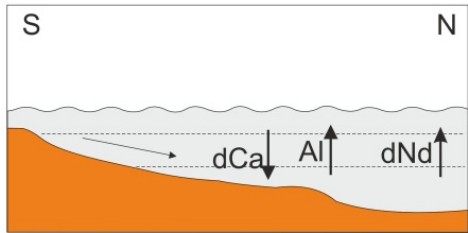

Stage 3: Deep water circulation

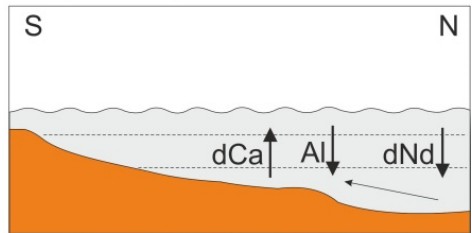

**Figure 11.** The three proposed models of the phosphate deposition.

Igneous provinces contain huge amounts of phosphorous which could be released during chemical weathering of igneous rocks during tectonic events in the Precambrian period [47]. Ref. [48] considered seafloor weathering along with continental weathering as first-order controls on phosphorous ocean productivity in the Phanerozoic oceans. Jordan in Campanian times witnessed an instability in basin architecture during the convergence process [26,27] which made it possible for submarine weathering or serpentinization to release phosphorous from the submarine igneous provinces.

As a second source that provided a countable amount of phosphorous into the ocean, an enhancement in the primary productivity due to the availability of nutrients in the ocean provided phosphorus through the degradation of organic matter in the deepest settings and the availability of bone fragments within sediments [3,5]. Bone fragments have been intensively observed in the section 1 samples and in sample HS-4 of section 3.

The effect of submarine weathering has been observed in a wide period at the north, while it was short in the south. A long period of high Nd isotope and low Fe, Al, Ti, and Al values indicated that the phosphatic-enriched ocean was characterized with submarine weathering of igneous provinces and a major decline in the hydrologic cycle, which could have occurred for two reasons: deep water was warm for a certain time in the Campanian period, during which the sea surface was relatively cold [22]; the architecture of the basin is the second factor as the shape of the Neo-Tethys during the convergence process was progressively changed to be restricted even to the hydrologic cycle.

### 5.3.3. Continental Weathering of Campanian

The long-term establishment of phosphate-rich intervals in section 3 is represented by a low δ43/42 Ca ratio, a high δ143/144 Nd ratio, and periodical changes in the terrigenous input. The skeletal grains were missed in this section. Coated and sometimes uncoated peloids supported with calcite cements were common in the samples. Both geochemical

and petrographical evidence supports the role of terrigeneous input and the weathering of the pre-existing continental igneous rocks in enriching the phosphate in the water column.

Weathering of the pre-existing continental igneous rocks can intensively enrich the phosphorous in the water column [8,9,49]. This can be enhanced in the warming periods, during which the hydrologic cycle is amplified in the Campanian period. Fe, Al, Ti, and Al were enriched along with the phosphorous ion in the water column, while low Ca was related to the weak demand on the skeletal grains in the phosphate component.

The land–sea configuration played a role in continental weathering and in the enrichment of phosphates in the water column. The basic assumption supported by many authors is that the southern part of Jordan was considered a proximal area of the basin [24,26]. The proximal area was supplied by a riverine system during the acceleration of the hydrologic cycle and huge amounts of terrigenous input were added into the basin during this time.

Fluctuations in the element association occurrences (Fe, Al, Ti, and Al) occurred in periodic events, as is a common process during seasonal changes in the strength of riverine systems. In contrast to this observation, the enrichment of phosphorous during submarine weathering appeared to be happen in bulk and be composed of a single event.

### 5.3.4. Deep-Water Circulation

The late Campanian period was characterized by the coldest period of the Late Cretaceous [22,23]. Based on the fact that the cooling went along with changes in global oceanic circulation towards a mode of deep-water formation [23], there was another period whereby a deep, cold-water body carried phosphates and other nutrients into the shallower part of the basin.

This event was observed in the top of section 1 (north) and in section 3 (south). Section 3 phosphate showed low Ca and Nd isotopes and low terrigenous indicators. This phosphate layer was composed of peloids with missing of skeletal fragments in its texture. The top phosphate layer at section 1 showed low terrigenous indicators, such as Al, Si, Ti, and Fe, but high Nd and Ca isotopes were enriched. Ca isotopes were enriched because of the abundance of skeletal fragments in this layer. There was evidence from the high Nd isotope that submarine weathering was affecting the northern part of Jordan. This effect did not appear in the south.

### 5.3.5. Middle Eocene Phosphate

The closure of Neo-Tethys during the middle Eocene led to a major regression in the eastern margin of the African Plate and a restriction of basins [17]. Initially, there was significant increase in Al, Fe, Si, and Ti in the carbonate-rich phosphate horizon. The $\delta 143/144$ Nd showed a slight increase and decline in the $\delta 43/42$ Ca isotope. During the main regression stage of the sea, phosphorous showed high values and was accompanied by high Al, Si, Fe, and Ti. The $\varepsilon$Nd and $\delta 43/42$ Ca isotope in their minimum values suggest that the restriction of the basin was established and the basin architecture allowed for terrigenous weathering through an acceleration of the hydrologic cycle in the middle Eocene time.

## 6. Conclusions

Phosphate was deposited throughout Jordan during the Campanian to Eocene periods. These deposits are varying due to differences in oceanographic and climatic conditions. Biostratigraphy, petrography, and geochemical investigation were performed on 79 samples from south, central, and north Jordan. Grainstone is the dominated facies in the phosphate samples, which is composed of skeletal fragments and peloids. There is a difference in the textural composition of phosphates between the south and north. Skeletal fragments along with the pellets appeared to be the main components of section 1, while sections 2 and 3 contained peloids with an absence of skeletal grains. This suggests that phosphates deposited in the inner parts of the shelf extend throughout Jordan. However, each phosphate deposition was affected significantly by its position in the shelf. At least three intervals of

high phosphorous values appeared in section 1 a showing variation in Ca and Nd isotopes and in the terrigenous inputs. The same has been observed in sections 2 and 3. Four periods of phosphate enrichment were proposed; these are the deep-water circulation at the early and late Campanian periods, interrupted with two periods of submarine and continental weathering. Deep-water circulation was initiated during the cooling in the Campanian period and was indicated by high phosphorous and Ca isotope components and a decline in the terrigenous indicators (Al, Si, Ti, and Fe). Submarine weathering was established during warmer deep-sea periods and indicated by a rising Nd isotope ratio, and many of the igneous provinces were subjected to the weathering. Continental weathering took place in the warmer periods due to the hydrologic cycle and the enhancement of terrigenous indicators (Al, Si, Ti, and Fe) in the ocean. The effect of the hydrologic cycle was at its maximum in the south in the Campanian and Eocene periods, as both represented shallower settings. Calcium isotopes suggest that the deposition of the apatite was mediated through inorganic processes, and the REE patterns indicate that this occurred in oxygenated marine water with few later diagenetic processes.

**Supplementary Materials:** The following supporting information can be downloaded at: https://www.mdpi.com/article/10.3390/app13031568/s1.

**Author Contributions:** Conceptualization, N.A.-J.; Methodology, M.A.-T.; Software, A.A.-R.; Formal analysis, M.A.-T.; Investigation, A.A.-R. and M.A.-T.; Writing—original draft, M.A. and A.A.-R.; Writing—review & editing, N.A.-J.; Project administration, M.A.; Funding acquisition, M.A. All authors have read and agreed to the published version of the manuscript.

**Funding:** This research was funded by the Scientific Research Fund, Ministry of Higher Education, Fund number BAS1/2/2019.

**Institutional Review Board Statement:** Not applicable.

**Informed Consent Statement:** Not applicable.

**Data Availability Statement:** The data presented in this study are in Supplementary Materials.

**Acknowledgments:** This research was funded by the Scientific Research Fund, Ministry of Higher Education, Fund number BAS1/2/2019. Authors thank Ali Delki and Tariq Qdaisat from the Jordanian phosphate company for help in sample collection.

**Conflicts of Interest:** The authors declare no conflict of interest.

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
