# Peer review of "Paleoenvironmental Study of the Late Cretaceous–Eocene Tethyan Sea Associated with Phosphorite Deposits in Jordan"

_applsci, doi:10.3390/app13031568_

Round 1

Reviewer 1 Report

Dear authors,

I would like to aknowledge the interest of this manuscript.  with helps to understand the factors that may affect the deposition and enrichement of P deposists and the potential implications they have for te climate research and the industrial prospection.

Just to comment some things about the figures 6-7-8. please consider in wich extend the smoothness of the elemental profile might be useful, since authors have not modeled that behaviour. It seems that authors could want to show a linear variation of the concentration i.e. along a single stratigraphic section

Fig. 9 is confusiong since  there are numerous colored lines overlaped. Is this figure really necessary? In my opinion it does not gives additional value to te text.

Pag.6 section 3 Materials and methods Element composition. MS-ICP-MS ???? Do you mean quadrupole ICP-MS , HR-ICPMS, MC ICP-MS? please provide more details about the digestion and measurement conditions (instrumental conditions, they may be eused two different MS for the measurements...).

Thank you very much

Regards

Author Response

Dear Reviewer,

Thank you for the opportunity to revise our manuscript for resubmission following outcome of you and your recommendations. We found the review to be highly helpful and they allowed us, as we believe, to improve on the quality of the manuscript. Overall, we took all suggested changes.

With reference to the your comments we would like to respond to each point in the table:

Comments

Just to comment some things about the figures 6-7-8. Please consider in which extend the smoothness of the elemental profile might be useful, since authors have not modeled that behavior. It seems that authors could want to show a linear variation of the concentration i.e. along a single stratigraphic section

We just plotted the values without using any smoothness of elemental profiles. This because we have few samples and trends are clear to certain extend.

Fig. 9 is confusiong since there are numerous colored lines overlaped. Is this figure really necessary? In my opinion it does not gives additional value to te text.

Figure 9 has modified in the new version to avoid the confusing. Fourteen curves were chosen to represent the RRE.

Pag.6 section 3 Materials and methods Element composition. MS-ICP-MS ???? Do you mean quadrupole ICP-MS , HR-ICPMS, MC ICP-MS? please provide more details about the digestion and measurement conditions (instrumental conditions, they may be eused two different MS for the measurements...).

We apologize for the mistake. The instrument has corrected in the manuscript. Detailed information added to the manuscript in the regards of the equipment.

With kind regards,

Yours sincerely,

Mohammad Alqudah

Reviewer 2 Report

The manuscript is good and may be accepted for publication.

Author Response

Dear Reviewer,

Thank you for the opportunity to revise our manuscript for resubmission following outcome of you and your recommendations. We found the review to be highly helpful and they allowed us, as we believe, to improve on the quality of the manuscript. Overall, we took all suggested changes.

With reference to the your comments we would like to respond to each point in the table:

Comments

 English language and style are fine/minor spell check required

Thank you for the comment, English has improved in the new version. All corrections are in track change mode.

With kind regards,

Yours sincerely,

Mohammad Alqudah

Reviewer 3 Report

The manuscript by Mohammad Alqudah et al. demonstrated the petrological, geochemical and biostratigraphical characteristics of four selected sections in Jordan to reveal the enrichment and deposition of massive phosphorite deposits in the southern margin of the Neo-Tethys of Jordan. The main contents of the paper include four aspects:

(1) The biostratigraphical study not only provided evidences (nannofossil marker species) to rebuild the ages of the section (Campanian and Eocene periods) and the paleo-climate information of the environment during phosphate precipitation (the enrichment of some ecological marker species indicate cold periods), but also disclosed the components of the phosphates (the skeletal fragments and peloids).

 (2) The geochemical studies of Al, Si, Ti and Fe contents and the Ca and Nd isotopes and their vertical variations not only reveal the existence of three intervals of phosphates, but also demonstrate the effects of terrigenous inputs and submarine weathering during warmer deep-sea periods.

(3) The spatial and vertical distribution of phosphorous layers in the 4 sections are summarized.

(4) The influence factors for the enrichment of phosphates are discussed, including the benefit oceanographic setting and sedimentary environments, the submarine and continental weathering during the warmer periods, the sources from the input of the terrigenous materials and the hydrologic cycle, etc.

The study of this paper helps to clarify the uncertainties of the origin of the huge phosphate deposits in this area and its controlling factors which helps to improve the exploration and development of the phosphates.

Because of the widespread of the phosphate-rich strata in Campanian-Eocene, especially the large economic interest of the massive phosphate deposits in the southern Neo-Tethys, any progress about these deposits is of great significance for revealing the mineral resource potential in this area, so the significance of this study is great. Most contents of the paper are well presented, so it is worth for publishing. However, there are also a lot of problems in this paper, including many small mistakes of grammar and expressions in the context and the figures (see in the attached modification version), some sections of paper are not well organized and described, and a major drawback is the authors have not clearly and fully summarized the conclusion which is not correspondent to the discussion. A moderate reversion is needed before publication, and the following are some specific comments:

1. For Section 2, it is short of some important information.

The distribution of the 4 sections and their primary geological backgrounds should be introduced in this section since the following context is absent of this kind of information.

2. For Section 3, it needed to be reorganized as following:

3.1 Materials:

(1) Total samples and their distribution in the 4 sections; It would be better to make a table to list the samples and their basic features, and further information about the purpose for the collection of these samples and different analyses items.

(2) The lithological units in each section should be introduced in more detail from bottom to the top. A correlation of lithological units in the 4 sections is absent.

3.2 Methods

One more problem: the information of Section 4 (the Al-Shidiyya section) is missing.

The contents of the Fig. 3 in this section are not well described, i.e., the lacks of data references and the introduction of the other two profiles (Well XD2 and HD1), the comparison of the three profiles.

3. For Section 4 (Results):

(1) The Section 4.1 should pay more attention to the distribution and occurrence of the biomarkers, a table list of the identified fossils in each section is absent.

(2) In Section 4.2, the division of facies should be given, which is based on different contents of the lithological components and textures. Then, to be more logical, lithological characteristics of the 4 sections should be introduced one by one.

(3) The section 4.3 can be subdivided into major and trace element compositions, Ca and Nd isotopes, and the primitive data table should be presented in the paper or as attached files.

4. Section 5 (discussion) and 6 (conclusions) are not well suited with each other, and the data of the study are not well severed for the discussion, as a result, the conclusions of this paper are not logically supported by the data and discussion.

For Section 5, it needs to be reorganized and re-written to state the controlling factors for the enrichment of phosphates in the 4 sections. At least the following questions should be answered:

5.1 The distribution and correlation of phosphates in the 4 sections;

5.2 The different sedimentary environments and deposition of the phosphates in the north and south Jordan;

5.3 The controlling factors for the formation of phosphates, which can be further subdivided into: the oceanographic settings; the sources of the phosphates (the terrigenous input, and materials by submarine weathering); the deep-sea circulations; the effect of the hydrologic cycle etc.  

For section 6, the conclusions also need to be re-written, and the following contents should be clearly stated: the distribution and enrichment of phosphates in the 4 section; the main characteristics of the phosphate enriched layers; the controlling factors of the enrichment of phosphates (the oceanography settings, the source materials, the sedimentary environments and the deep-sea currents).

Conclusions and suggestions:

The paper is of some significance to reveal the origination of phosphate resources in the southern margin of Neo-Tethys. The study provided a series of useful data, especially the geochemical data of the phosphate-bearing series. Most contents of the article are well organized and presented, so it is qualified for publication on the journal. Anyway, there are some mistakes and even fatal errors in the context. Moderate modification is need before publication.

Author Response

Dear

Thank you for the opportunity to revise our manuscript for resubmission following outcome of you and your recommendations. We found the review to be highly helpful and they allowed us, as we believe, to improve on the quality of the manuscript. Overall, we took all suggested changes .

With reference to the your comments we would like to respond as follows:

Comments

  However, there are also a lot of problems in this paper, including many small mistakes of grammar and expressions in the context and the figures (see in the attached modification version),

Grammar and expressions are revised as per the reviewer instructions. One thing only that upwelling issue did not cover well. That because we are looking beyond the main factor which is the upwelling. However, authors added statements in the discussion describe why we concentrated on climatic and oceanographic factors.  

 Some sections of paper are not well organized and described, and a major drawback is the authors have not clearly and fully summarized the conclusion which is not correspondent to the discussion.

Sections revised and we tried to organize the discussion and conclusion sections.

 For Section 2, it is short of some important information.

The distribution of the 4 sections and their primary geological backgrounds should be introduced in this section since the following context is absent of this kind of information.

Geological background and distribution of sections and formations were added to the new version.

2. For Section 3, it needed to be reorganized as following:

3.1 Materials:

(1) Total samples and their distribution in the 4 sections; It would be better to make a table to list the samples and their basic features, and further information about the purpose for the collection of these samples and different analyses items.

A table were added describing the sample numbers and the lithological description of each samples and the type of analysis on each sample.

(2) The lithological units in each section should be introduced in more detail from bottom to the top. A correlation of lithological units in the 4 sections is absent.

Correlation has made and added to text of section 3

3.2 Methods

One more problem: the information of Section 4 (the Al-Shidiyya section) is missing.

Al shidiyya has distinguished in sub title section 4

The contents of the Fig. 3 in this section are not well described, i.e., the lacks of data references and the introduction of the other two profiles (Well XD2 and HD1), the comparison of the three profiles.

We added the number sections into the sections in figure 3.

3. For Section 4 (Results):

(1) The Section 4.1 should pay more attention to the distribution and occurrence of the biomarkers, a table list of the identified fossils in each section is absent.

We added the distribution charts of the marker species attached to figure 4. 

(2) In Section 4.2, the division of facies should be given, which is based on different contents of the lithological components and textures. Then, to be more logical, lithological characteristics of the 4 sections should be introduced one by one.

An introductory statements explains the division of facies in the samples were added.

(3) The section 4.3 can be subdivided into major and trace element compositions, Ca and Nd isotopes, and the primitive data table should be presented in the paper or as attached files.

We added a supplementary data in excel format includes all the major, trace and isotopic data.

4. Section 5 (discussion) and 6 (conclusions) are not well suited with each other, and the data of the study are not well severed for the discussion, as a result, the conclusions of this paper are not logically supported by the data and discussion.

We Added statements to the discussion and conclusions.

for Section 5, it needs to be reorganized and re-written to state the controlling factors for the enrichment of phosphates in the 4 sections. At least the following questions should be answered:

5.1 The distribution and correlation of phosphates in the 4 sections;

The discussion has been modified and the section 5.1 The distribution and correlation of phosphates in the 4 sections has added to the new version.

5.2 The different sedimentary environments and deposition of the phosphates in the north and south Jordan;

Also, this section has been added into the discussion.

5.3 The controlling factors for the formation of phosphates, which can be further subdivided into: the oceanographic settings; the sources of the phosphates (the terrigenous input, and materials by submarine weathering); the deep-sea circulations; the effect of the hydrologic cycle etc.  

Sub-sections were added under the main title: 5.3 The controlling factors for the formation of phosphates

For section 6, the conclusions also need to be re-written, and the following contents should be clearly stated: the distribution and enrichment of phosphates in the 4 section; the main characteristics of the phosphate enriched layers; the controlling factors of the enrichment of phosphates (the oceanography settings, the source materials, the sedimentary environments and the deep-sea currents).

Conclusion has been modified based on the reviewer comment.

With kind regards,

Yours sincerely,

Mohammad Alqudah
